

**PeerJ Hubs**
Published on behalf of

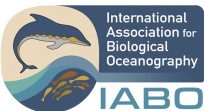
International Association for Biological Oceanography
IABO

# Findings of new phytoplankton species in the Barents Sea as a consequence of global climate changes

Pavel Makarevich[1], Viktor Larionov[1], Anatoliy Oleinik[1] and Pavel Vashchenko[2]

[1] Department of Plankton, Murmansk Marine Biological Institute of Russian Academy of Science, Murmansk, Murmansk Region, Russia
[2] Department of Environmental Engineering, Murmansk Marine Biological Institute of Russian Academy of Science, Murmansk, Murmansk Region, Russia

## ABSTRACT

Over the past few decades, the Earth's climate has been characterized by a stable increase in temperature, which in many regions leads to a change in the composition of flora and fauna. A striking manifestation of this process is the appearance in ecological communities of new, uncharacteristic for them, species of animals and plants. One of the most productive and at the same time the most vulnerable in this respect are the marine ecosystems of the Arctic. This article is devoted to the analysis of findings of vagrant phytoplankton species in the Barents Sea, a body of water experiencing especially rapid warming due to an increase in the volume and temperature of incoming Atlantic water. For the first time, fundamental questions are considered: how widely these species are distributed over the Barents Sea area, and in what seasons do they reach high levels of abundance. The material for the present work was planktonic collections made during expedition surveys of 2007–2019 in different seasons throughout the Barents Sea. The water samples were taken using a rosette Niskin bottle sampler. The plankton net with a 29 μm mesh size was applied for filtering. The obtained material was processed according to standard hydrobiological methods and followed by subsequent microscopy for taxonomic identification of organisms and cell counting. The results of our observations show that vagrant microplankton species do not create a stable population that persists throughout the annual development cycle. Their major presence is noted in the autumn-winter period, the smallest in the summer. The distribution of invaders is strictly tied to warm jets of currents, while the weakening of the inflow of Atlantic water masses deep into the Barents Sea from the west is a limiting factor for their penetration into its eastern part. The southwestern and western parts of the basin are characterized by the most significant number of floristic finds; from here, towards the north and east, their number decreases. It can be concluded that at present the proportion of vagrant species in the Barents Sea, both in species diversity and in the total biomass of the algocenosis, is insignificant. They do not change the structure of the community as a whole, and their presence does not have any negative impact on the ecosystem of the Barents Sea pelagic. However, at this stage of research, it is too early to predict the environmental consequences of the phenomenon under study. Given the growing number of recorded cases of finds of species uncharacteristic for the Arctic, there is a possibility that this process may disrupt the biological stability of the ecosystem and even lead to its destabilization.

Corresponding author
Pavel Vashchenko,
pavelvashenko@gmail.com

# INTRODUCTION

One of the most important fields of bio-oceanology in recent decades has been the study of modern climatic changes and their consequences for marine ecosystems (*Comiso & Hall, 2014*; *Dong et al., 2020*). The manifestation of this process is that the increased temperature might make it possible for new species to colonize new areas after being transported with the currents (*Reid et al., 2007*).

This phenomenon is especially important in Arctic pelagic ecosystems, which are the most productive and, at the same time, the most vulnerable from an ecological point of view (*Fermandez, Kaiser & Vestergaard, 2014*). It is at high latitudes that natural variations in climatic parameters reach their maximum extent, in particular, water temperatures in the Arctic Ocean are rising faster than in other parts of the globe, and this trend is expected to intensify in the coming century (*IPCC, 2013*). At the same time, even relatively small changes in the natural environment can go beyond the adaptive capacity of some species of flora and fauna, which will inevitably lead to serious disturbances, both in individual communities and in the ecosystem as a whole (*Fermandez, Kaiser & Vestergaard, 2014*).

In the Barents Sea, the described process is named by specialists as "Atlantification" (*Årthun et al., 2012*; *Bagøien et al., 2018*). Since the 1980s, under the influence of global climate change, this water body has been undergoing a rapid warming trend (*Ingvaldsen & Loeng, 2009*). This is due to changes in the hydrological parameters of the Barents Sea as a result of the increased volume and temperature of incoming Atlantic water (*Neukermans, Oziel & Babin, 2018*). Oceanic currents and increased water temperature directly contribute to the establishment of vagrant species in new water areas for them (*Occhipinti-Ambrogi, 2007*; *Sorte, Williams & Zerebecki, 2010*). However, only a fraction of them can adapt to their new environments (*Crooks & Soulé, 1999*; *Mack, Simberloff & Lonsdale, 2000*). Examples include northward and the eastward expansion of the ranges of Barents Sea crab populations: snow crab *Chionoecetes opilio* and king crab *Paralithodes camtschaticus* Stachowiczetal2002 (*Starikov et al., 2015*; *Spiridonov & Zalota, 2017*; *Zalota, Spiridonov & Vedenin, 2018*). The same climatic changes are thought to result in a shift to the north and east of the sea of the boundaries separating warm-water and cold-water Decapoda complexes (*Zimina et al., 2015*) and boreal and arctic fish species communities (*Fossheim et al., 2015*; *Bagøien et al., 2018*).

But most of all, the increased inflow of Atlantic waters and warming affect the structure of pelagic algocenoses, causing changes in their taxonomic composition due to the penetration of new species of tropical and tropical-boreal origin (*Oleinik, 2014*; *Ardyna & Arrigo, 2020*; *Ardyna et al., 2020*; *Wang et al., 2018*). A number of our articles have detailed findings of vagrant microplankton species in the Barents Sea (*Oleinik, 2014*; *Makarevich & Oleinik, 2017*; *Makarevich & Oleinik, 2020*). However, these publications lack information on how widely these species spread over the water body, how long they remain in the Barents Sea pelagic zone throughout the year, and in which seasons they reach high abundance levels.

The purpose of this article was to analyze the materials obtained to answer these questions to estimate the scale of possible changes in the structure of phytoplankton communities. Its results are of paramount importance for predicting negative consequences for the Arctic marine ecosystems as a whole.

## MATERIALS & METHODS

The material for the present work was planktonic collections made during expedition surveys of 2007–2019 in different seasons throughout the Barents Sea (*Makarevich & Oleinik, 2020*). The scheme of sampling stations is shown in the figure (Fig. 1). Dates, time, sampling coordinates, depths at the points of hydrobiological work are given in the appendix (Appendix S1).

   The water for phytoplankton samples were taken using a rosette Niskin bottle sampler (Multi Water Sampler ROSETTE HydroBios MWS-12, Altenholz, Germany). Water samples were taken from standard hydrobiological sampling horizons –0, 5, 10, 25, 50, 100 m and the bottom layer (*Dybern, Ackefors & Elmgren, 1976*). Additionally, a net with a filter cone made of gas with a mesh size of 29 μm was used to catch the entire water layer. CTD profiles were performed at all sampling points. The obtained material was processed according to standard hydrobiological methods: samples of 1–2 L were concentrated using the reverse filtration method to a final volume of 4–5 ml; after that, they were fixed with a 40% formaldehyde solution, with final concentration 2–4% (*Dodson & Thomas, 1964*). Fixed net samples of phytoplankton were concentrated by sedimentation method (*Dybern, Ackefors & Elmgren, 1976*). After settling the sample in the dark, the water over the settled precipitate was concentrated by dropping it with a thin glass siphon with an upturned end to a volume of 2 ml.

   For taxonomic identification of organisms and cell counting, a Palmer-Maloney counting chamber was used (*Karlsen, Cusack & Beensen, 2010*). Microscopy study was performed under an AxioImager D1 light microscope (Carl Zeiss, Jena, Germany) at 400x magnification. The names of species and systematic groups, as well as phytogeographic characteristics of microalgae, are given according to the nomenclature from electronic sources (*AlgaeBase, 2022*; *WoRMS, 2022*).

   The dynamics of temperature anomalies in the waters of the Barents Sea is given on the basis of year-round observations on the standard oceanographic section ''Kola Meridian'' (*PINRO, 2023*). The standard oceanographic section ''Kola Meridian'' is located in the Barents Sea to the north of the Kola Bay along 33° 30′E from 69° 30′to 77° 00′N and it consists of 16 stations (Fig. 1). Its length is 450 miles. Depth at stations varies from 150 m to 310 m and averages 245 m (Appendix S2). The stations of the section are located in the area of following waters distribution: The Coastal and Main branches of the Murmansk and Central branches of the North Cape currents (*Ozhigin et al., 2011*), *i.e.,* the section crosses all the main water masses of the Barents Sea in terms of its genesis–coastal, Atlantic and Arctic.

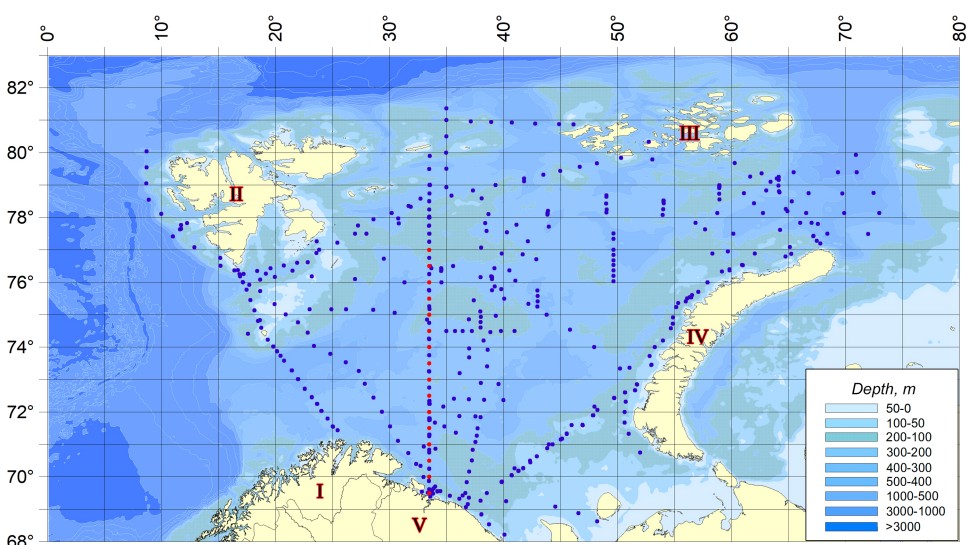

**Figure 1** **Sampling stations location.** Red dots indicate the stations related to the Kola Meridian section.

# RESULTS

In the period from 2007 to 2019, microalgae that had not previously been found in this water body, or their findings were considered questionable, were found in the Barents Sea. The list of species is given in Table 1. It includes 17 representatives of phytoplankton, of which 16 are dinoflagellates (Class Dinophyceae), and 1 is a diatom alga *Proboscia indica* (Class Bacillariophyceae). See Appendix S2.

Figure 2 shows the scheme demonstrating the distribution of vagrant species detection sites in the studied water area. The dense location of sampling stations, repeatability of surveys at the same points in different years, and coverage of almost the entire area of the Barents Sea allow to reliably draw borders of areas with different degrees of their occurrence.

Most of these invaders are common components of pelagic algocenoses of the seas of the North Atlantic Basin. The diatom *Proboscia indica* is known from the Norwegian and North Seas (*Nehring, 1998*), characterized as a thermophilic species, subtropical and boreal (*Hendey, 1964*). The boreal species *Amphidoma caudata*, the tropical-boreal *Corythodinium diploconus*, *Dinophysis hastata*, *Mesoporos perforatus*, *Pseudophalacroma nasutum*, *Oxytoxum caudatum*, *Podolampas palmipes*, and the tropical *Heterodinium milneri* are widely distributed in coastal waters of Britain and Norway (*Okolodkov Yu, 2000*). The boreal species *Protoperidinium laticeps* was described from the waters of West Greenland (*Grøntved & Seidenfaden, 1938*) and subsequently found in the Norwegian and Baffin Seas, temperate and subtropical regions of the Northeast Atlantic (*Okolodkov Yu, 2000*). The tropical-boreal *Pyrophacus horologicum* has been noted in the Norwegian, White, and Baltic Seas (*Okolodkov Yu, 2000*; *Hällfos, 2004*). The dinoflagellate *Spatulodinium pseudonoctiluca*, also of tropical-boreal origin, has a wide range: it is registered in the Northern and Baltic Seas, as well as in the Kara Sea and the Arctic Ocean (*Kiselev,*

**Table 1  List of pelagic microalgae species first observed in the Barents Sea.**

| Species | Area | Year/Month | Max N, cells/l (year/month; area) |
|---|---|---|---|
| *Amphidoma caudata* Halldal | $A_1$ | 2012/XI; 2013/XI | 25 (2012/XI) |
| *Ceratium strictum* Kofoid | $A_1$, $B_2$ | 2014/VI; 2015/VII,XI; 2019/XI | 20 (2015/VII; $A_1$) |
| *Corythodinium diploconus* Taylor | $A_1$ | 2012/XI; 2013/XI | less than 10 cells identified |
| *Dinophysis hastata* Stein | $A_1$ | 2013/XI | less than 10 cells identified |
| *Dinophysis ovata* Claparede et Lachmann | $A_1$, $B_2$ | 2013/XI; 2014/VI; 2015/XI; 2016/IV | less than 10 cells identified |
| *Gotoius mutsuensis* Matsuoka | $A_1$ | 2014/VI | less than 10 cells identified |
| *Heterodinium milneri* Kofoid | $A_1$ | 2013/XI | less than 10 cells identified |
| *Mesoporos perforatus* Lillick | $A_1$, $B_2$ | 2013/XI | 53 ($A_1$) |
| *Oxytoxum caudatum* Schiller | $A_1$, $A_2$, $B_1$, $B_2$, $B_3$, $B_4$ | 2007/VIII,IX; 2010/IX,X; 2012/XI; 2013/XI; 2014/VI; 2015/XI; 2016/IV; 2017/XII; 2019/XI | 400 (2013/XI; $A_1$) 300 (2012/XI; 2015/XI; $A_1$) 55 (2016/IV; $B_4$) 200 (2016/IV; $B_2$) |
| *Podolampas palmipes* Stein | $A_1$ | 2013/XI; 2016/IV; 2017/XII; 2018/I; 2019/XI | less than 10 cells identified |
| *Proboscia indica* Hernandez-Becerril | $A_1$ | 2016/IV | less than 10 cells identified |
| *Protoperidinium brochii* Balech | $A_1$ | 2013/XI | less than 10 cells identified |
| *Protoperidinium laticeps* Balech | $A_1$ | 2014/VI; 2019/XI | 98 (2014/VI) |
| *Protoperidinium thulesense* (Balech) Balech | $A_1$ | 2012/VI | less than 10 cells identified |
| *Pseudophalacroma nasutum* Jörgensen | $A_1$ | 2013/XI | less than 10 cells identified |
| *Pyrophacus horologicum* Stein | $A_1$, $B_2$ | 2013/XI | less than 10 cells identified |
| *Spatulodinium pseudonoctiluca* Cachon et Cachon | $A_1$ | 2015/VII | less than 10 cells identified |

*1950*; *Dodge, 1982*; *Druzhkov & Makarevich, 1999*; *Wasmund et al., 2015*). The species *Protoperidinium thulesense* is characterized as Pan-Arctic (*Okolodkov, 1996*): it was observed in the White and Kara Seas, as well as in the Japan and boreal zone of the Pacific Ocean (*Abé, 1981*; *Konovalova, 1998*; *Matsuoka et al., 2006*). Another group of microalgae: boreal *Ceratium strictum*, tropical-boreal *Dinophysis ovata* and *Protoperidinium brochii*, and *Gotoius mutsuensis* of unidentified origin are now reliably known only from materials from the Black Sea and the Mediterranean (*Kiselev, 1950*; *Gómez, 2003*; *Krakhmal'niy, 2011*).

A comparison of the selected sites in terms of the number of finds of vagrant species demonstrates the unconditional leadership of area $A_1$, in which all the vagrant species were found and the maximum number of their registrations was observed (Fig. 2). Here also the greatest values of the numbers reached by several organisms are marked. The second place is occupied by area $B_2$ vagrant phytoplankters. Only one microalga, *Oxytoxum caudatum*, was found in the water area of the other sites.

This species deserves special attention. It is present in the pelagic zone for almost the entire period of studies, in all seasons, and throughout the water areas of regular and attenuated occupation. Its populations reach concentrations an order of magnitude higher

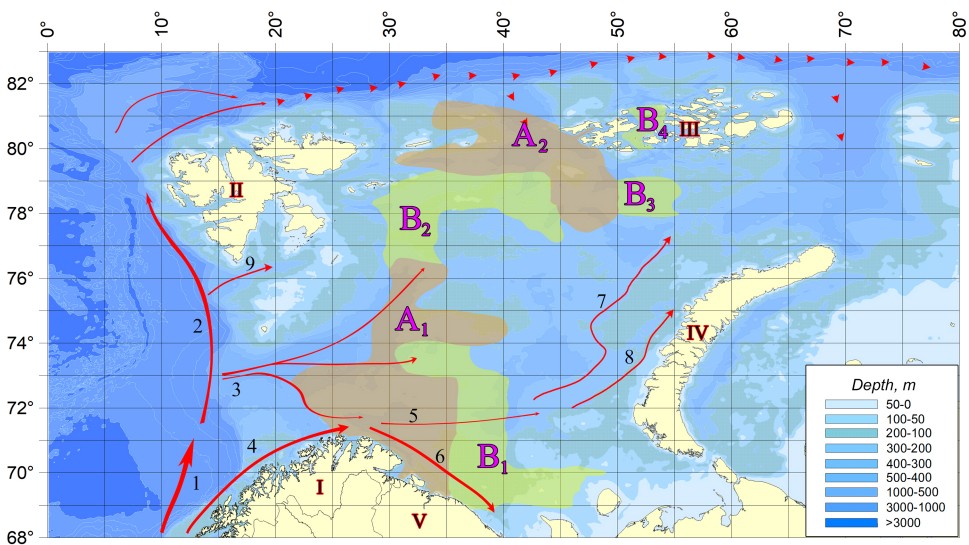

**Figure 2  The distribution of vagrant species detection sites in the studied water area.** (A) In areas marked on the map with letter A ($A_1$ and $A_2$)—a zone of regular infestation—the vagrant species were registered at every observation. In areas marked with B ($B_1$, $B_2$, $B_3$, and $B_4$)—the zone of weakened occupation—only in individual years. In the rest of the water body, species not typical of the Barents Sea pelagic algoflora were not found. I, Scandinavian Peninsula; II, Svalbard; III, Franz Jozef Land; IV, Novaya Zemlya; V, Kola Peninsula; 1, Norwegian Atlantic Current; 2, Fram Strait Branch; 3, Barents Sea Branch; 4, Norwegian Coastal Current; 5, Murmansk Current; 6, Murmansk costal Current. 7, West Branch of Novaya Zemlya current; 8, Coastal Branch of Novaya Zemlya current; 9, South Svalbard Current; dashed line, Atlantic water bottom current.

than those of other vagrant organisms. It can be assumed that *Oxytoxum caudatum* is the only representative of algoflora not typical of the Barents Sea, which has been adapting to new environmental conditions in the initial stage.

The distribution of the number of finds by year is as follows. The maximum number of encounters—11—is characterized by 2013, followed by 2014 (five cases), 2012, 2015, 2016, and 2019 (four registrations each), 2017 (two findings), 2007, 2010, and 2018 (one encounter each). A similar comparison of the number of omnivore sightings by month shows this pattern. The highest number of them (21) is in November, the other seasons are much less rich: June (six cases), April (four), July, September and December (two), January, August, and October (one) (see Table 1). During the other periods of the year, no vagrant species of microalgae were detected in the Barents Sea pelagic zone. However, it should be taken into account, that in February–March, under-ice vegetation occurs in all Arctic seas (*Ardyna & Arrigo, 2020*), and in May, a peak of spring phytoplankton bloom forms near the ice edge during the period of ice melting (*Perrette et al., 2011*); during these phases of the annual successional cycle, diatoms absolutely dominate in the composition of alcogeoses, and the proportion of dinoflagellates is extremely low (*Makarevich, Druzhkova & Larionov, 2012*).

It is also important to note that, based on the results of many years of research, it was found that in the Barents Sea pelagic, according to the phytogeographic characteristics,

approximately 40% of microalgae taxa are arcto-boreal species, 30% are cosmopolitan and 20 are boreal (*Makarevich & Druzhkova, 2010*). As part of the new finds, most organisms are representatives of the tropical-boreal and boreal algoflora, and a small number are tropical and bipolar. Previously, no species of tropical-boreal and tropical origin were recorded in the Barents Sea (*Matishov et al., 2000*; *Makarevich & Druzhkova, 2010*).

## DISCUSSION

As already noted, the main indicator of the dynamics of climatic factors is seawater temperature. We have at our disposal a multiyear series of year-round observations of hydrological parameters, in particular temperature, on the standard oceanographic transect "Kola Meridian" (Fig. 3). The series of observations are given from the site of Polar branch of the FSBSI "VNIRO" PINRO named after N.M. Knipovich. Their analysis indicates a clear increase in the advection of warm waters from the Atlantic to the Barents Sea in recent decades, which is the main reason for the appearance of vagrant species in the Barents Sea waters. The second way of transfer of vagrant organisms is anthropogenic activity, but for phytoplankton representatives in the seas of the Arctic basin, reliable cases of such introduction are currently unknown (*Stachowicz et al., 2002*; *Padilla & Williams, 2004*).

The inflow of Atlantic water masses into the Barents Sea occurs due to constant (non-periodic) currents, which together form a relatively stable circulation system within the reservoir (Fig. 2). These currents determine the general distribution of water masses in the Barents Sea water area and its water exchange with the adjacent areas (*Potanin, Denisov & Ershtadt, 1985*). On the western border, this water exchange is carried out with the Norwegian and Greenland Seas. Through the largest strait, between the island of Medvezhiy and the mainland. The largest strait between Medvezhiy and the mainland (Nordkapp), water flows from the Norwegian Sea to the Barents Sea through two currents– the Norwegian Atlantic Current and the Norwegian Coastal Current. Through another strait on the western border, between South Cape Island (Sørkapp–Norwegian name; Svalbard archipelago) and Bear Island, the South Svalbard Current enters the Barents Sea from the Norwegian Sea. The Norwegian Atlantic Current in the Barents Sea area is divided into several streams, passing further in the northern and eastern directions (*Matishov, Matishov & Moiseev, 2009*). The Northern Branch, following the Nadezhda Trough, divides into smaller streams that move northward to the west of the Perseus Plateau and eastward between the Perseus Plateau and the Central Bank (*Loeng, 1991*). Another part of the Northern Branch, deviating westward, forms a flow along the western edge of the Medvezhiyski Trough, directed into the Norwegian Sea (*Gawarkiewicz & Plueddemann, 1995*). Norwegian Coastal Current extends eastward and appears in the Demidovsky Trough and above the Central Upland (*Boytsov, 2006*). The Nordkapp South Current runs deep into the Barents Sea and divides into the Murmansk Current and the Murmansk Coastal Current, which flows along the northern and southern slopes of the Murman Rise in an easterly direction (*Terzieva, 1992*). On the northern boundary of the

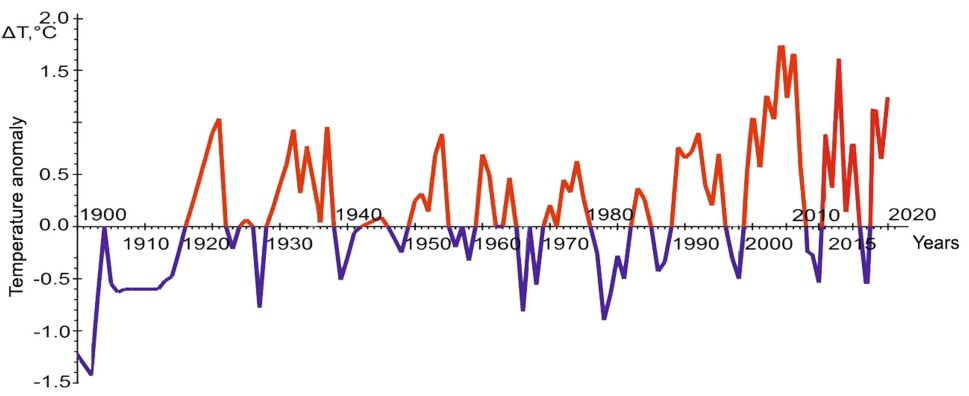

**Figure 3** The graph of temperature anomalies in the oceanographic transect "Kola Meridian". Figure based on (*Matishov, Matishov & Moiseev, 2009*), with additional data from *PINRO, 2023*.

reservoir, a complex system of surface currents is used to exchange water with the adjacent Arctic Ocean water area.

The analysis of Fig. 2 reveals a complete coincidence of the current directions (three streams of Atlantic water entering the Barents Sea) and the areas of regular settlement. Area $A_1$ is directly influenced by Barents Sea Branch of the Norwegian Atlantic Current. In areas $B_1$ and $B_2$, located near its borders, the speed of these currents is significantly reduced.

Area $A_2$ is the area affected by deep Atlantic water masses from the north, decreasing by the South Spitsbergen Current, which decreases in the nearby areas $B_3$ and $B_4$ (as well as in area $B_2$). In the eastern part of the Barents Sea, although there is an inflow of warm waters with jets of the Novaya Zemlya current, it is already too weak, and therefore invaders are not found in this area. Thus, there is a clear connection: the weakening of the currents leads to a decrease in the number of finds of vagrant species up to their complete disappearance. As a result, a completely natural situation is observed, when the richest in the number of vagrant species and registered encounters in area $A_1$, in which all 17 species are found. Area $B_2$, containing five representatives of vagrant algoflora, is under the influence of two less strong flows, and only one species of *Oxytoxum caudatum* is found in the other selected areas. Characteristically, this microalga reaches maximum concentrations (300–400 cells/l) in area $A_1$, being present throughout the studied water area during the entire period of research.

It should also be noted that the indicated abundance values of *Oxytoxum caudatum* were observed only in November, and lower, but comparable to the data (50–200 cells/l) in April–May (Table 1), and only on one depth level 200 m. At the same time, in the winter season (November–December), this species was often included in the composition of the dominant forms of the phytoplankton community (against the background of low general taxonomic diversity and low concentrations of organisms), and in some cases it was the only dominant (Appendix S2). Among the other algae reaching relatively high abundance levels (more than 20 cells/l), these were recorded in one month of the year, also predominantly in November (Table 1). At the same time, the rarest (single) findings were

recorded in June (*Gotoius mutsuensis*), July (*Spatulodinium pseudonoctiluca*), and also in November (*Heterodinium milneri, Pseudophalacroma nasutum*).

It should be emphasized that during the entire period of our research, the phytoplankton community in each hydrological season was represented by a microalgae complex, a set of dominants, and quantitative characteristics standard for a particular vegetation phase (*Makarevich, Druzhkova & Larionov, 2012*). The indicators of introduced species development were relatively stable from year to year, and the observed intra-annual fluctuations in the number of individual species corresponded to seasonal levels of abundance of algocenoses (*Hopes & Mock, 2015*). Thus, in no area do vagrant species create a stable population that would persist for a long period (at least one stage of the annual successional cycle). There is only their temporary presence in the Barents Sea, with the highest concentrations being recorded in the autumn-winter season when active growth has already stopped, and the lowest in summer. This fact can most likely be explained by the fact that it is November when the maximum volume of Atlantic water enters the water body (*Ingvaldsen, Asplin & Loeng, 2004a*; *Ingvaldsen, Asplin & Loeng, 2004b*).

The comparison of the number of the omnivore finds in different years shows that their maximum number is in 2013, the years 2012 and 2014–2016, as well as 2019, are less rich in the number of encounters, the periods from 2007 to 2011, 2017 and 2018 are represented only by isolated cases. The graph of temperature anomalies in the "Kola Meridian" oceanographic transect (Fig. 3) shows that the interval from 2012 to 2016 is characterized by high positive values of this indicator (with a peak in 2013), while the previous and subsequent periods are negative. This relationship strongly indicates that increased water temperature is a necessary condition for the adaptation of warm-water species in the Arctic basin, in particular, in the Barents Sea.

The process of organism's introduction includes a donor region, a vector, a corridor, a recipient region, a candidate species, and many factors that impede this process. All of these parameters are important in understanding the success or failure of a non-native species. However, characteristics of the recipient area, such as low species diversity, climate change, which are often considered to facilitate the process of introduction into terrestrial ecosystems, may be of much less importance in the marine environment; on the contrary, the role of the vector and the corridor can be underestimated (*Boudouresque & Verlaque, 2012*).

## CONCLUSIONS

Overall, the data presented suggest that the distribution of vagrant microphytoplankton species in the Barents Sea is tied to the warm currents coming from the Atlantic Ocean. The weakening force of these water masses penetrating deep into the Barents Sea from the west turns out to be the main factor limiting the presence of vagrant species in the eastern part of the reservoir. As a result, the southwestern and western parts of the water area are characterized by the greatest number of floristic findings, and further to the north and east the number of such species decreases. The maximum diversity of this group of organisms is confined to the warmest years when the inflow of Atlantic water masses increases.
At present, the share of pelagic microalgae new to the Barents Sea in the total taxonomic diversity is insignificant. There is no mass development of them in any season of the year, they do not form high biomasses and do not change the community structure as a whole. As a result, their appearance does not lead to the destabilization of planktonic algocenoses themselves and does not have any negative impact on other components of the Barents Sea pelagic ecosystems.

Nevertheless, at this stage of research, it is too early to predict the ecological consequences of the occurrence of vagrant microalgae species in Arctic waters. Moreover, taking into account the increasing number of recorded finds of phytoplankton representatives uncharacteristic for the Arctic, we can assume that if the positive temperature trend persists, the process of occupation will increase its intensity. In this case, negative consequences are possible: changes in the community structure, oppression of native species, and decrease in biological stability of pelagic ecosystems.

### Funding
This work was supported by the Russian Academy of Science, state project topic "Features of the organization of Arctic plankton communities in modern climate change" No. 9-21-01 (1.6.16) 121091600105-4 (16.09.2021). The funders had no role in study design, data collection and analysis, decision to publish, or preparation of the manuscript.

### Grant Disclosures
The following grant information was disclosed by the authors:
The Russian Academy of Science, Features of the Organization of Arctic Plankton Communities in Modern Climate Change: 9-21-01 (1.6.16) 121091600105-4 (16.09.2021).

### Competing Interests
The authors declare there are no competing interests.

### Author Contributions
- Pavel Makarevich conceived and designed the experiments, performed the experiments, analyzed the data, authored or reviewed drafts of the article, and approved the final draft.
- Viktor Larionov conceived and designed the experiments, performed the experiments, analyzed the data, prepared figures and/or tables, authored or reviewed drafts of the article, and approved the final draft.
- Anatoliy Oleinik conceived and designed the experiments, performed the experiments, analyzed the data, authored or reviewed drafts of the article, and approved the final draft.
- Pavel Vashchenko analyzed the data, prepared figures and/or tables, authored or reviewed drafts of the article, and approved the final draft.

## Data Availability

The data is available in the Table and Figures.

## Supplemental Information

Supplemental information for this article can be found online at http://dx.doi.org/10.7717/peerj.15472#supplemental-information.

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
