# Peer review of "Findings of new phytoplankton species in the Barents Sea as a consequence of global climate changes"

_PeerJ, doi:10.7717/peerj.15472_

## Round 0.1 · original submission · Major Revisions

Both referees recommend some expansion, clarifications, and corrections to the writing; and improvements to the Figures and Table. Please revise and respond to these suggestions.

Reviewer 1 ·

Basic reporting

The English is mostly fine, but I have detected some uncommon wordings:
Line 73: ‘development’ of alien species should perhaps rather be ‘establishment’ of alien species.
Line 139: ‘… is characterized as bipolar…’. I expect bipolar to be something found in Arctic and Antarctica, while her the authors perhaps mean ‘Pan-Arctic’?
Line 146: ‘pests’. Are the alien species all really ‘pests’?
Line 238: ‘… on one horizon – at depth of 200m.’ is more English as ‘… on one depth level 200m.’
Line 243: ‘…when active vegetation has already stopped…’ is more English as ‘…when active growth has already stopped…’?
The references appears to me to be quite comprehensive and relevant.
The structure of the article conform to standards suggested by PeerJ.
Figures 1 and 3 lack information on CTD station positions of the “Kola Meridian” section (temperature anomalies shown in Figure 2), and phytoplankton sampling positions. I would suggest to add the Kola Meridian section to Figure 1, and phytoplankton sampling stations to Figure 3.
Line 146: Figure 3 should be referred to here when mentioning area A1 and A2, implying Figure 3 and Figure 2 should switch number.
Figure 1 legend contains some unfamiliar names of currents. I would recommend 2 – Fram Strait Branch; 3 – Barents Sea Branch. Perhaps also 6 – Cola Coastal Current.
I don’t find any information about raw data availability.
The submission is self-contained if figures are changed as suggested, and description of currents in discussion is simplified.

Experimental design

The submission is original primary research within the aims and scope of the journal (research article in Biological Sciences).
Research questions are clearly defined and meaningful.
Investigation appears rigorous.
Methods described to such a degree it can be replicated.

Validity of the findings

Interesting results, but they are only briefly presented in Figure 3 and Table 1.
Underlying data have not been provided, so their soundness cannot be assessed.
Conclusions appear appropriate, but could be better connected to the currents shown in Figure 1.
Lines 191-222: Contains far too detailed description of currents, with place names not indicated in any figure. The Currents indicated in Figure 1 are sufficient to explain the findings in regions A and B in Figure 3. Regions B are downstream of Atlantic Water inflow to regions A.
Line 227: ‘South Spitsbergen Current’ is not mentioned earlier in the manuscript. My interpretation of region A2, is a connection to the deep inflow of Atlantic Water from the north. The authors should address the validity of this suggestion.

Additional comments

Interesting results, but some times unclear presentation.

Reviewer 2 ·

Basic reporting

no comment

Experimental design

methods should be better described, for example sampling positions, depths, count method (see additional comments).

Validity of the findings

The results/findings are not sufficiently described (see additional comments).

Additional comments

This short article describes the observations of alien phytoplankton species in the Barents Sea during sampling surveys from 2007 to 2019.The authors discuss these observations in light of the observed warming of the Arctic and Barents Sea region during latter decades and the Atlantification of the Barents Sea water masses. This is a timely and interesting topic for sure and the manuscript is well written with clear language. My main point of concern is that both methods and results should be better described, for example sampling positions, depths, count method and for results from which depths, number of cells counted, what were the dominating species and so on. See the specific comments
below. I think these issued should be addressed by the authors before acceptance.

line 19: biocenoses is correct term but not commonly in use anymore, maybe consider call it for example "ecological communities"?
line 32: was the reverse filtration performed by using the plankton net? Was the volume reduced to 4-5 mL in the cod end of the net? Please clearify
line 34: organisms
line 34: "chambers of various volumes"...which technique was used, not utermohl chambers? reference?
line 55-56: more precicely the increased temperature might make it possible for new species to colonize new areas after being transported with the currents?
line 104: reference for the hypochlorite method?
line 114: are the sampling/detection sites supposed to be shown in Fig. 1? The figure text mention only the currents.
line 146: the areas A1, A2 and so on are shown in Fig. 3, so the authors should refer to it here
line 157-163: the results are from Table 1 so the authors should refer to it
line 164: what do the authors mean by subglacial vegetation? is it ice algae? If so, then February-March seems very early for the Arctic. The peak og the ice algae bloom is
normally in May? At least add a reference for this.
line 178-179: should this be in the results section? also describe the Kola Meridian, where was the temperature measured?, give a reference to description of the
time series.
line 186-207: it would be good to include at least the most important currents of the eastern part of the Barent Sea in fig. 3.

Table 1: What does it mean when the last column is empty? that only one cell was identified? Both microscopy count method and the results should be better described. If these data are from previously published results then put another column in the table with the reference. In the original counts how many cells were counted to have proper statistics? Position of sampling points should be indicated in the map. What depths were sampled? It would be interesting to know what others species were found and the total phytoplankton cell density. Figure 1 and 3: can easily be combined into one in their present form. But I argue that the sampling stations or transects should be shown on a map.

---

## Round 0.2 · Minor Revisions

Thank you for a well-revised paper. Can you replace the word "invasive", typically used for human-introduced species, to "vagrants" or similar, as these are natural movements of species? Please also take the opportunity to check the text, figures etc for any small amendments you feel improve the paper.

Reviewer 1 ·

Basic reporting

I am happy with the changes done. Perhaps change ‘alien species’ to ‘vagrant species’. This has been suggested to me as a better wording.

Experimental design

No additional comments.

Validity of the findings

I am happy with the changes made on names of currents. Description is clearer now.

Additional comments

Abstract is very long. Consider removing most of the Methods part.

Reviewer 2 ·

Basic reporting

no comment

Experimental design

no comment

Validity of the findings

no comment

Additional comments

The authors have made a good effort to improve the manuscript based on the review comments. I have no further comments.

---

## Round 0.3 · accepted · Accept

Thank you for the final revisions.